# Consumer Perception of Freshness and Volatile Composition of Fresh Gilthead Seabream and Seabass in Active Packaging with and without CO_2_-Emitting Pads

**DOI:** 10.3390/foods12030505

**Published:** 2023-01-21

**Authors:** Evangelia Nanou, Mado Kotsiri, Dimitra Kogiannou, Maria Katsouli, Kriton Grigorakis

**Affiliations:** 1Institute of Marine Biology, Biotechnology and Aquaculture, Hellenic Centre for Marine Research, 46.7 km Athens-Sounio Ave., 19013 Attiki, Greece; 2Laboratory of Food Chemistry and Technology, School of Chemical Engineering, National TechnicalUniversity of Athens (NTUA), 15780 Athens, Greece

**Keywords:** consumer acceptance, CO_2_-emitter, sensory freshness, fish shelf life, modified atmosphere, volatile compounds

## Abstract

Active packaging with CO_2_-emitters (pads) has recently been used for shelf-life extension of fresh fish. The aim of this study was to identify consumer attitudes towards fresh fish packaging, to examine whether Greek consumers prefer active packaging with pad over active packaging without pad, to investigate any perceived differences in the sensory freshness of the fish, and to relate consumer perception to volatile composition of fish fillets. In total, 274 consumers participated in the study which included freshness sensory evaluation of gilthead seabream (*Sparus aurata*) and seabass (*Dicentrarchuslabrax*), whole-gutted and filleted, raw and cooked, at high quality and at the end of high-quality shelf-life. Samples were packed under modified atmosphere either with a pad or without. Results showed that consumers preferred packages with pads, especially at the end of high quality shelf-life. They perceived raw samples packed with a pad to be fresher and closer to the ideal product, and also had a higher purchase intention towards them. Cooked samples were not perceived differently. Consumers’ perception was in accordance with the GC-MS findings in the volatile compounds that function as freshness or spoilage indicators. Most participants were positive towards fresh fish packaging although they usually buy unpacked fresh fish. Our results suggest that active packaging with CO_2_ emitters contribute to freshness preservation and that it has a positive potential in the Greek market.

## 1. Introduction

Fish is an important food commodity due to its high nutritional value; its global production in 2020 reached a record 214 million tonnes and is foreseen to increase by a further 15% by 2030 [1]. Therefore, it is important for the industry to adopt new sustainable technologies and successful market approaches that will cover these demands. One of the major limitations during fish production and processing is the highly perishable nature of its products. Thus, shelf-life extension and quality assessment of fish during the whole process is of utmost importance [2]. In this context, the search for new methods to preserve fish during storage and for freshness evaluation have caught the interest of scientists.

The use of modified atmosphere is widely used for fresh fish storage as a means to enhance shelf-life [3,4]. In addition, CO_2_-emitting pads have been used in the recent years to maintain the CO_2_ concentration in the packaging and to prevent spoilage [5,6,7,8,9]. Consumer purchase behavior is a complex mechanism that involves various factors concerning the product, such as its origin, preservation method, physical properties, and availability; and factors concerning the consumer, such as attitudes, sensory perception, preference and eating habits [10,11,12,13]. Within these factors, perceived quality is one of the most important ones when buying fish [14], and freshness is a very important quality parameter [15].

However, there is no data available on the use of active packaging with CO_2_-emitters in the acceptance of fish products by the consumers. Examining the attitudes of the consumers towards active packaging with pads would give an important insight into the utility and potential of this method as a means to limit food waste by extending the shelf-life of the product. Furthermore, involving consumers early on in the new product development is proven to be the key for success in the market place [16]. A very recent study on co-creation of fish packaging with consumers has pointed out the importance of consumers’ perception and opinions during packaging design [17].

Thus, the aim of this research was to examine consumer preference for active packaging of gilthead seabream (*Sparus aurata*) and seabass (*Dicentrarchuslabrax*) with CO_2_-emitters and to identify consumer attitudes towards fresh fish packaging. Furthermore, we aimed to investigate potentially perceived differences in the sensory properties of the products from the consumer’s point of view in raw and cooked samples. Within this framework, we also attempted to relate the consumer’s perception of freshness with the volatile composition of the fish in different packaging (active packaging with and without pad).

## 2. Materials and Methods

### 2.1. Sample Preparation and Storage

Gilthead seabream and seabass of cage-farming origin were obtained from AVRAMAR S.A. (Athens, Greece). Standard commercial procedures were used for scale and gut removal prior to the filleting process. Afterwards, samples of whole-gutted fish (300–400 g) and skin-on fillets (100–140 g) were packed with ice and shipped, under refrigeration, to the processing plant (SelectFish S.A., Athens, Greece), where they were immediately packaged under modified atmosphere, MAP (40% CO_2_, 40% N_2_, 20% O_2_). Packaging took place within 48 hours after slaughter, which is a customary time interval in the industry, and packaging conditions were the those commonly preferred by the fish production company in question. Half of the packages, hereafter named MAP-PAD, additionally contained a CO_2_-emitting pad (90 × 255 mm, maximum absorbency: 100 mL, maximum CO_2_ production: 128 mL for the whole-gutted fish, and 80 × 130 mm, maximum absorbency: 35 mL, maximum CO_2_ production: 120 mL for the fillets). Pads were kindly provided by McAirlaid’s (Berlingerode, Germany). Thereafter, samples were immediately transported to the Hellenic Center for Marine Research (HCMR) under cold chain, and kept in refrigeration at 2.5 °C until used in the consumer test or analyzed. This temperature was chosen based on previous findings on optimum storage under MAP and MAP-PAD conditions for the same species [18].

### 2.2. Questionnaire

Two questionnaires were created for the evaluation of consumer preferences and their perception of fish freshness; one with regard to the whole fish and a second one regarding the fillets. Whole fish were evaluated as raw samples; fillets were evaluated both as raw and cooked samples. Both questionnaires consisted of questions about the perception of fish freshness and consumer preference, in the first part. The last part of the questionnaire consisted of demographic questions and questions regarding attitudes towards fish consumption and packaging. Table 1 presents in detail the questions asked in the two questionnaires and the corresponding scales used. Scales were converted to numeric data from 1 to 5 or 1 to 3 according to the points of the respective scale. The more positive the rating was, the higher the score.

### 2.3. Experimental Design and Conducting of Consumer Test

In total, 274 consumers took part in the study (44% male; 56% female) with an age range of 21 to 62 years (mean age: 40 ± 10 years). Participants were mostly HCMR employees and were recruited on the basis of their interest and availability to participate in the study. They were invited to the study via an e-mail sent to all personnel. The only prerequisite for participation was not to detest fish. Details on the distribution of the consumers over the test sessions can be seen in Figure 1.

The test took place in the Sensory Lab of the Institute of Marine Biology, Biotechnology and Aquaculture at HCMR in Anavyssos. A schematic overview of the consumer test experimental design can be seen in Figure 1. Four sessions, on different days, took place between October and November 2021. Sessions were run from 10.30 a.m. to 16.30 p.m., and complied with the institute’s COVID-19 protocol for hygiene and social distancing.

Two freshness time points were chosen to test the samples: one at the point of highest quality (HQ) during its shelf-life, i.e., the beginning of storage (immediately after transportation to HCMR); and one at the end of the high-quality shelf-life (End). The end of high-quality was defined based on previous experience of MAP storage of the studied species [18]. High quality coincides with bacterial lag phase period, which was determined at around 8 to 9 days from slaughtering. We chose the end of high-quality, instead of the end of shelf-life time point, in order to keep a market-oriented approach where the test samples would still be in a proper condition to be sold. Four samples were evaluated in each session by each consumer; two seabream packages, one with a pad (MAP-PAD) and one without a pad (MAP), and two respective seabass packages. On the day of the fillets’ evaluation, consumers additionally evaluated four cooked samples which came from the respective packaging of the raw samples.

Participants were asked to first open the package and then start evaluating successively the odor, appearance and texture of the raw sample. Respectively, for the cooked samples, they evaluated their odor, taste and texture. No information was given to the participants regarding the packaging and the presence or absence of a pad.

Raw samples were presented to the participants after having been kept for 15 minutes at room temperature, so that odor and appearance be evaluated in a more representative and systematic manner. During preparation of the cooked samples, pieces of about 20 g (2 × 2 × 1 cm) were cut from the dorsal part of the fillet, that is, about two pieces from each fillet, in order to have homogenous samples. The samples were placed in ceramic containers covered with aluminum foil and cooked in the oven at 110 °C for 20 minutes. At the end of the cooking process, the samples were placed in a thermal chamber at 60 °C by the time of serving and for a maximum of 30 minutes.

All samples, raw and cooked, were presented in randomized order and were blind coded with 3-digit numbers in order to account for first order and carry-over effects [19]. Water and spittoons were available for palate cleansing after the tasting of each cooked sample.

### 2.4. Volatile Compounds Analysis

Samples for volatile compounds (VOCs) analysis were isolated from the packed fillets of the consumer study. The VOCs were determined by headspace SPME-GC/MS analysis according to Katsouli et al. [18] based on a modified procedure of Parlapani et al. [20]. Specifically, volatiles were extracted by homogenizing 5 g of minced fish muscle with 4 mL of saturated saline and incubated at 40 °C for 15 min. The SPME fiber (50/30UM DVB/CARBOXEN-PD) was exposed to the headspace for an additional 40 min, under the same conditions. The headspace was then analyzed using gas chromatography–mass spectrometry (Agilent Technologies, Santa Clara, CA, USA) and the separation was achieved on an Agilent DB-WAX GC Column (30 m 0.25 mm, coated with a 0.25 μm film thickness). 4-methyl-1-pentanol was used as an internal standard; and the identification of the compounds was based on comparing MS data with those of reference compounds and by MS data obtained from the NIST library (NIST/EPA/NIH Mass Spectral Library with Search Program, software version 2.0f) and by semi-quantitative analysis using the method of internal standard. All samples were analyzed in triplicate.

### 2.5. Chemical Freshness Analysis

The freshness assessment of gilthead seabream and seabass fillets was based on an ATP breakdown products analysis (K-values index). K-value was determined as described in Grigorakis et al. [21] method. K-value definition was based on the ratios of the concentrations of the end products of ATP breakdown (Hx and Ino, respectively) and those of the intermediate compounds (ATP metabolites). Concentrations of individual compounds were quantified by comparison of their chromatogram areas with the standard curves of adenosine 5’-triphosphate (ATP), adenosine 5’-diphosphate (ADP), adenosine 5’-monophosphate (AMP), inosine 5’-monophosphate (IMP), inosine (Ino), and hypoxanthine (Hx) (Sigma-Aldrich, St. Louis, MI, USA).

### 2.6. Microbiological Analysis and MAP Gas Changes

The evaluation of microbial quality deterioration of gutted and filleted fish samples was based on enumeration of total viable count (TVC), *Pseudomonas* spp., *Enterobacteriaceae* spp., and H_2_S-producing bacteria (i.e., *Shewanella* spp.) following the method described by Katsouli et al. [18]. TVC was grown on plate count agar (PCA, Merck, Darmstadt, Germany), *Pseudomonas* spp. on Cetrimide agar (CFC, Merck, Darmstadt, Germany), whereas H_2_S-producing bacteria were grown on Iron Agar (Iron agar, Merck, Darmstadt, Germany); their colonies were enumerated after incubation at 25 °C for 72 h, 48 h, and 48 h respectively. *Enterobacteriaceae* spp. was grown on violet red bile glucose agar (VRBG, Merck, Darmstadt, Germany) and incubated at 37 °C for 24 h.

The changes of the gas headspace (CO_2_, O_2_) into the packaging during storage were determined using the CheckMate 9900 O_2_/CO_2_ meter (PBI Dansensor, Rinsted, Denmark).

### 2.7. Data Analysis

Data analysis was focused on the investigation of differences between MAP and MAP-PAD samples at the same time point (high-quality or end of high-quality). Thus, data were analyzed for each species, seabream or seabass, and each processing type, gutted fish or fillet, separately within each time point (HQ or End). The latter was decided in order to follow the study design of testing different time points on different days.

Preference data for the raw samples are reported as frequencies and were analyzed by applying a binomial test. Sensory freshness data were checked for normality and were found not to be normal (Shapiro-Wilk test, *p* < 0.05 and visual observation of histograms and P-Plots). Thus, the Mann–Whitney U test was run to test differences in the scores of the attributes between samples with and without a pad, at high quality and at the end of high-quality time points. Attributes were treated as dependent variables, while presence or absence of a pad was treated as an independent variable, and data were analyzed at high quality (HQ) and end of high quality (End) time points, separately. The chi-square test was used for categorical variables, such as presence or absence of liquid in the packaging, mucus on the skin, and coherency of the tissue, as well as consumer attitudes and frequency of consumption.

Volatile data were analyzed by means of two-way ANOVA considering the presence or absence of pad and high-quality or end of high-quality time points as main effects, and including their interaction effect. Volatiles for which a significant effect (*p* < 0.05) or a tendency for significance (*p* < 0.1) was shown, as well as significant freshness perception attributes from the consumer study as supplementary data, were subjected to principal component analysis (PCA). Chemical freshness data (K-values) were analyzed by a Mann–Whitney U test, as described above for sensory freshness data, because data were not normally distributed.

All analyses were based on the 5 % significance level (α = 0.05). IBM SPSS Statistics Version 28 (IBM Corporation, Armonk, New York, NY, USA) was used for the binomial, Mann–Whitney U, chi-square tests, as well as for the two-way ANOVA. PCA was run in XLSTAT 2022 (Addinsoft, New York, NY, USA).

## 3. Results and Discussion

### 3.1. Consumer Attitudes

Fish consumption frequency and consumer attitudes towards fish packaging are presented in Table 2. Results show that almost all participants of the study (99%) consumed fish on a weekly or monthly basis. Furthermore, they usually purchased fresh fish, and unpackaged fish in particular. These results are in accordance with previous research conducted in Greece that indicated that at least 77% of the participants consumed fish once per week and that almost 80% preferred to eat fresh fish [22]. In another study, it was reported that the vast majority of the studied population (88% men and 91% women) consumed fish at least once a month [23]. Moreover, among eight European countries, Greece has been rated second after Portugal in the consumption frequency of fish, especially fresh fish, though not among the first in the total per capita consumption [24].

Regarding their attitudes and beliefs, more than half (59%) of the respondents believe that packaged fish is more convenient than unpackaged fish, 38% believe that packaged fresh fish is safer, while 50% believe that it is less fresh than unpacked fresh fish. The latter finding is also reflected in their preference for buying fresh fish that is not packed (88%). The above results stress the complexity of fish quality which involves both product characteristics but also consumer perceptions [10,11,12]. Perception is dependent on consumers’ knowledge and cognition but also on their emotions, beliefs and general socio-culture heritage [10,25,26]. Product attributes that have been identified to play a role in fish consumption and purchase behavior are, among others: origin, preservation method, fish freshness, convenience, availability, safety, and price [11,13,27]. Furthermore, consumer perception can also influence the potential of a product in the market [10]; thus, attitudes towards fish consumption, such as healthiness aspects and taste [12], but also perceived image and trust in the end product [28], may be of great importance.

Finally, the participants of our study were asked if they would prefer to buy a packaging with a pad over one without a pad. More than half of them replied positively, while almost 1/3 remained indifferent. However, only a small percentage of the participants were clearly negative towards the packaging with a pad (Table 2). Thus, taking into consideration the positive and indifferent answers as well, we could infer that there is a generally favorable attitude towards the pads. In contrast to the openness to new technology of these particular consumers, previous research on Greek consumers has reported their preference for more traditional packaging methods and methods with technology that is probably easier to grasp, such as vacuum—as the majority of those consumers reported that they were not aware of the function of the different packaging types [22]. This stresses the importance of raising awareness among consumers, and public policies should orientate towards this goal.

In our study, the proportion of consumers that were negative about the use of pads may prefer to be able to see any kind of spoilage in the food that comes with storage time, and may be afraid that the pad would mask such changes. This observation is based on oral communication with the participants who replied negatively, after the end of the test session. This is in accordance with the literature that reports the importance of trust in the end-product [28] and the fact that although consumers seek better quality of packed food, they are not fond of technologies that make them feel deceived [29].

### 3.2. Consumer Preference and Freshness Evaluation of Raw Samples

Seabream and seabass, as whole-gutted fish and as fillets, respectively, and packaged under modified atmosphere, were used in this study. For each species, participants were asked, after opening the two types of packaging, which sample they preferred, MAP or MAP-PAD ones, without giving any information about the samples during testing. Herein it must be noted that there were no direct comparisons made between highest quality and end of high-quality shelf-life, but only between the different packaging conditions (MAP-PAD or MAP) at the same time point. As can be seen in Table 3, whole seabream with a pad was preferred over its counterpart at the end of high-quality shelf-life (End). Regarding fillets, the samples with a pad at the end of high-quality shelf-life were the most preferred for both species. Seabass fillets with a pad at the high quality shelf-life time point (HQ) were preferred as well, while a similar tendency can be observed for seabream fillets. These findings indicate that the pad has a positive effect on the preference of the consumers, especially at the end of high-quality shelf-life. It is noteworthy that different total numbers of consumers per session can be observed due to the distinct days on which the test sessions took place (Figure 1). Moreover, the difference in the total numbers of consumers can also be attributed to consumers who gave no answer in the preference question and therefore did not appear in the frequency count.

To our knowledge, these are the first data that prove the acceptability among consumers of such a packaging designed to improve shelf-life and reduce food waste in fish products. Perceived quality is one of the most important factors that influence food choice, especially in fish. Fish freshness, among other factors such as safety, nutritional content, and physical properties, significantly affects perceived quality [11,14].

Regarding the sensory freshness evaluation of whole-gutted gilthead seabream (Appendix A), consumers rated the MAP-PAD samples at the end of high-quality shelf-life as fresher in odor (*p* = 0.004) and overall (*p* = 0.011) compared to MAP ones. Furthermore, this was reflected in their willingness to buy (*p* = 0.018) and closeness of the product to the ideal of packaged fish (*p* = 0.009). As for the whole-gutted seabass, MAP-PAD samples were not different from MAP samples at any time point for any of the attributes (Appendix A). For both species, with respect to the liquid in the packaging, the chi-square test analysis showed that, regardless of the freshness stage, MAP-PAD conditions exhibited less liquid (*p* < 0.05) than the respective MAP samples (Appendix A). This result was expected since the liquid that leaks from the fish tissue is absorbed by the pad. Mucus presence did not differ significantly between groups at any of the two freshness stages (Appendix A).

With regard to fillets (Appendix A), gilthead seabream did not exhibit any significant difference between MAP-PAD and MAP samples at either time point. Nevertheless, seabass fillets with a pad (MAP-PAD) were perceived as fresher in odor (*p* = 0.030) and overall (*p* = 0.028), while consumers were more willing to buy them (*p* = 0.045) and perceived them as closer to the ideal product of packaged fish (*p* = 0.010) at the end of high-quality shelf-life compared to their counterparts without a pad (MAP). Similarly to the whole-gutted fish, the presence of liquid was significantly higher (*p* < 0.05) in MAP fillets than in MAP-PAD ones (Appendix A), regardless of the species or freshness stage; this indicates a significant impact of the pad in absorbing sample-excreted liquid, besides its main role as gas releaser.

A significantly (*p* < 0.05) higher rated fresh odor, overall freshness perception, purchase intention and proximity to the ideal product for the MAP-PAD samples in whole seabream and filleted seabass indicated that, besides the self-explained relation between freshness perception and purchase intention, perceived freshness seems mainly to be attributed to the perception of fresh odor, more than the other sensory attributes that did not differ significantly between different packaging. Previous research has pointed out the complexity of freshness and quality evaluation of fish by consumers [13], while elsewhere it has been reported that consumers feel uncertain or find it difficult to evaluate freshness [30].

Apart from fresh odor, the flesh tended to be perceived as brighter white and firmer in MAP-PAD gilthead seabream (*p* = 0.091) and seabass (*p* = 0.071) fillets at the end of high quality comparing to their MAP counterparts. Appearance affects perceived quality, and in turn consumer decision on the point of purchase [31,32,33]. Thus, liquid in all tested samples, and to some extent flesh color in the fillets, may have been easier for the consumers of this study to evaluate. Consumers may be more familiar with these appearance attributes given that they are more distinctive in the packed fish and fillets during purchase. On the other hand, skin in the fillet, or eyes and skin in the whole fish, may be more difficult for the consumer to evaluate before opening the package.

The use of consumers in the evaluation of fish fillet freshness was applied in a recent study [31], in which consumer acceptance or rejection at different time points of storage was investigated, in order to define the sensory acceptability limit on fish stored in refrigeration for days. Questions posed were about liking on a 9-point scale and about consumption and purchase intention on a binary (yes/no) scale. The results proved the value of using consumers in freshness evaluations in a multivariate approach together with, but not limited to, analytical tools. Our study is the second one that has used consumers to test the freshness of real fish products and the first one that has used them to test the potential of MAP-PAD packaging on the market.

### 3.3. Consumer Evaluation and Acceptance of Cooked Samples

Consumers tasted cooked samples of gilthead seabream and seabass fillets and evaluated them with regards to some sensory freshness characteristics and to their liking. Both species did not show any significant differences either as to the sensory perception or as to the degree of liking, when MAP-PAD and PAD samples were compared between highest quality and the end of high-quality shelf-life time points (Appendix A). Firmness of the cooked tissue was also evaluated and analyzed by the chi-square test; however, no differences were observed in any of the species at any time-point (Appendix A). The fact that no perceived difference was observed in freshness characteristics may be attributed to the cooking process that may mask any perceivable differences present in the raw fillets [34]. Nevertheless, overall freshness tended (*p* = 0.095) to be scored higher in seabass fillets at the high quality time points for the MAP-PAD compared to the MAP samples.

This is the first study to investigate consumers’ perceived freshness of cooked fish samples, since previous research with untrained [35] or semi-trained [36] consumers has focused on the sensory profiles of different cooked fish species, different fish origin (farmed vs. wild) [37] or culinary preparation effects [38]. Recently, Alexi et al. [39] looked into the effect of cooking on the perceived freshness of seabream fillets in different shelf-life time-points and processing, albeit using a trained sensory panel. Unlike our results, they found that differences in freshness were still perceivable after cooking. However, it should be noted that a trained panel differs from consumers in the capacity of detecting and describing differences [40].

### 3.4. Volatile Compounds and Relationship to Consumer Evaluation of Freshness

In gilthead seabream fillets, a total of 38 volatile compounds including 9 alcohols, 6 ketones, 7 aldehydes, 4 aromatic hydrocarbons, 6 alkanes, 3 esters, 2 terpenes and 1 imine, were identified and quantified. A total number of 14 out of these VOCs differed significantly or showed a tendency for difference (*p* < 0.1, Appendix A) between samples, and these were included in the PCA biplot (Figure 2) to visualize correlations between VOCs and samples, as well as between the consumers’ perception of freshness. In seabass, the same volatiles, apart from 2,3-octanedione that was not detected, were identified and quantified, and 15 of those (*p* < 0.1, Appendix A) were used in the PCA biplot (Figure 3). 

F1 and F2 explained 89.97% and 89.82% of the total variation in the data for seabream and seabass, respectively, indicating that they sufficiently represented most of the data (F1 = 79.05% and 62.46%, and F2 = 14.92% and 27.36%, for seabream and seabass, respectively). For both species, F1 separates the samples according to their freshness time-point (Figure 2 and Figure 3). Samples at the end of high quality (End) can be seen on the positive side of the axis, while samples at the high quality (HQ) time point are placed at the negative side of the axis. However, in both cases, this separation is clearer for the MAP samples. Furthermore, in seabream, MAP packaging had a major contribution in F1, while for MAP-PAD this was in F2. In seabass, F2 separates the samples according to the presence or absence of pad, with MAP samples on the positive side of F2 and MAP-PAD on the negative one.

In seabream, most of the significant VOCs are on the positive side of the F1 axis and correlate with the MAP and MAP-PAD samples at the end of high-quality time point, such as 2-penten-1-ol, hexanal, Z-4-heptenal, 1-octen-3-ol, 2-butanone-3-hydroxy, 2,3-pentanedione. Additionally, PAD samples at the high quality (HQ) time point are correlated with ethyl acetate, octanal, 1-octanol and heptanal. This indicates that HQ samples probably have a rather neutral odor compared to the ones at the end of high quality, since they have lower concentrations of the aforementioned volatiles.

In seabass, most of the significant VOCs, such as 2,3-pentanedione and 1-penten-3-ol, correlate strongly with MAP samples at the end of high quality. Some of the VOCs are on the positive side of F2 correlating with MAP at the high quality time point, such as octanal. Decanal is strongly correlated with MAP-PAD samples at the high quality time point and with consumers’ freshness perception attributes.

For both species, the consumers’ perception of freshness variables (fresh odor, overall freshness, purchase intention, and proximity to the ideal product) are positively correlated with each other, as expected. These attributes, although not significant in seabream, were chosen to be included as supplementary data in the PCA because they are indicative of consumer freshness perception as implied by their significant effect on seabass fillets. Furthermore, perception attributes are strongly positively correlated with MAP-PAD samples at the high quality time point, relating also to consumers’ tendency to prefer these samples (see Table 3). In seabream, they are negatively correlated with both MAP and MAP-PAD at the end of high quality, while in seabass, the negative correlation affects only MAP fillets at the end of high-quality shelf-life. This is again in accordance with the preference of the consumers who chose MAP-PAD seabass fillets, at both high quality and end of high-quality shelf-life time points, preferring these to MAP samples.

Aldehydes, such as decanal and octanal, are present in the same quadrants of the PCA with seabass fillets at the high quality shelf-life time point, and indeed are characteristic of them (Figure 3) as also reported in previous research [41,42]. Specifically, decanal is negatively correlated with hexanal, 2,3-pentanedione, and 1-penten-3-ol, which are associated with fish rancidity [43]; in addition, octanal is positively correlated with MAP samples at the high quality time point and with consumers’ perceived freshness, suggesting their potential as freshness markers of seabass.

The presence of Z-4-heptenal, hexanal, 2,3-pentanedione, 3-hydroxy-2-butanone, 1-octen-3-ol, and 2-penten-1-ol in the 4th quadrant of the PCA plot of seabream fillets and respectively, hexanal, 2,3-pentanedione, and 1-penten-3-ol in the 1st quadrant of the PCA plot of seabass fillets, indicates that these VOCs are most closely associated with the late days of storage (End) and mainly with MAP samples (Figure 2 and Figure 3); this is also justified by the fact that these compounds are involved with oxidation of polyunsaturated fatty acids [43]. In confirmation, these compounds were indeed found to increase (*p* < 0.05) or tend to increase their individual concentrations with storage (Figure 4 and Figure 5, Appendix A). These compounds are hereafter referred to as target compounds. Moreover, similar findings have previously been found for them in a study including various time storage intervals [18].

Z-4-heptenal, found in this study to be a target volatile for gilthead seabream, is responsible for off-flavor in fish. It can be either derived from lipid oxidation of n-3 PUFA or produced via 2,6-nonadienal that is produced by the action of 12-lipoxygenase on eicosapentaenoic acid (EPA, 20:5n-3) [44,45], and has already been identified in gilthead seabream [46]. Hexanal, found here to be an important freshness determinant for both species, has already been identified in European seabass and proposed as a spoilage marker by Leduc et al. [41]. Another characteristic target compound in both species associated with fish rancidity is 2,3-pentanedione, which is formed during storage [43], as is the case for 3-hydroxy-2-butanone [47]. The 1-octen-3-ol is a noteworthy contributor to off-flavors due to its low odor score, and has been reported to derive from the oxidation of arachidonic acid by 12-lipoxygenase [43,44]. As for 1-penten-3-ol, it is a characteristic alcohol detected in many fish species [47,48,49] and is produced during storage from EPA following reactions with 15-lipoxygenase and hydroperoxide lyases [48]. Furthermore, it is described as giving a fish-like odor to fish products during storage [50]. Moreover, 2-penten-1-ol has been recently characterized as a potential spoilage marker for red seabream due to its increasing levels during storage [51] and has also been identified as a potent odorant in raw sardine [52].

Finally, among all conditions, the highest total average VOC content was observed in packaging without pad at the end of high quality, as expected (Appendix A). The target compounds at the end of high quality presented the highest amounts in MAP packaging conditions rather than in MAP-PAD (Figure 4 and Figure 5); this clearly indicates that the presence of pad can have a positive impact on the freshness of gilthead seabream and seabass fillets by retaining lower storage-eluted volatiles in the samples.

### 3.5. Chemical and Microbial Freshness

K-values increased over the storage period (Appendix A), and values at the end of storage were similar to those previously reported for the same species when the fish approximated their acceptability limits [34,53,54]. However, the presence of pad did not have a significant effect on this index. Our results are in accordance with the recent literature on chemical and microbial assessment of fish freshness in active packaging [18]. Furthermore, we observe that K-values for fillets are higher than the corresponding values for whole fish. This indicates that fillets are more perishable than whole-gutted fish.

The use of CO_2_ emitters increased the CO_2_ concentration in the packaging headspace by the end of storage. In whole-gutted fish, CO2 concentration in MAP packages ranged from 19% to 22% at the end of high-quality storage, while in MAP-PAD packages, it was in the range 24–30%. The TVCs and H_2_S-producing bacteria were lower for most MAP-PAD samples compared to MAP samples (Appendix A). The TVC counts were found to remain below 7.0 log CFU g^−1^ which is, according to the literature, the acceptability limit for shelf-life based on total viable counts load [7,55,56]. In summary, the higher CO_2_ concentration in MAP-PAD seem effectively to have delayed the aerobic bacteria growth. Microbial findings confirm what the sensory and volatile compounds indicated for the effectiveness of the pad.

## 4. Conclusions

This is the first study to look into consumer perception for active packaging in fish using real-life products, both raw and cooked, in terms of preference, sensory freshness and attitudes. Consumers generally value the safety and convenience aspects of fish packaging, although a large proportion is still unconvinced about the utility of the packaging in freshness preservation. However, in theory, they are positive towards the addition of a pad that may boost the quality of packed fish. When raw fish or fillets were evaluated, consumers preferred MAP-PAD samples, especially at the end of high-quality shelf-life. This preference can be related to higher perceived freshness in terms of fresh odor and overall freshness perception, but also to the impact of the pad on the appearance of the sample by absorbing sample-excreted liquid. On the other hand, the presence of the pad did not affect consumers’ acceptance or perceived freshness of cooked samples.

Perceived freshness has been strongly related to volatiles that have been previously identified as indicators for fish freshness (octanal and decanal), whereas negatively correlated with volatiles (1-penten-3-ol, 2-penten-1-ol, hexanal, Z-4-heptenal, 1-octen-3-ol, 3-hydroxy-2-butanone, 2,3-pentanedione) that can be considered as spoilage indexes. The addition of CO_2_-emitting pads exhibited a positive impact on these aforementioned freshness indicators.

Furthermore, we demonstrated the importance of involving consumers in market research for the better understanding of their needs and perceptions regarding improved quality of fish products. Thus, conducting further research on active packaging and other new technologies with them could lead to a successful market approach that may contribute to extended shelf-life of products and, in turn, to the elimination of food waste.

## Figures and Tables

**Figure 1 foods-12-00505-f001:**
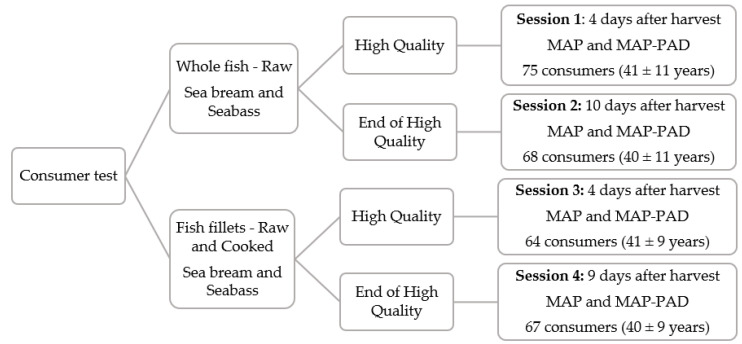
Schematic overview of the consumer test experimental design. MAP: modified atmosphere packaging, MAP-PAD: modified atmosphere packaging with CO2-emmiting pad.

**Figure 2 foods-12-00505-f002:**
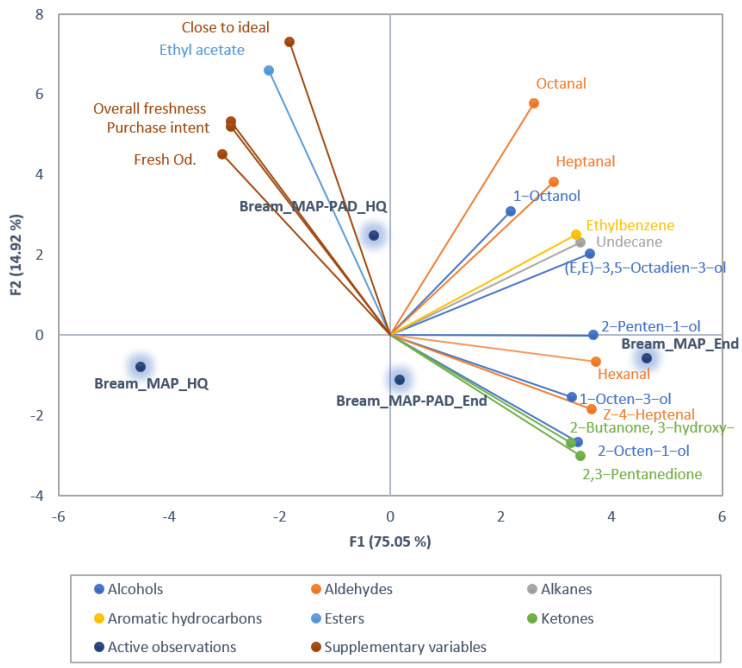
Principal component analysis (PCA) biplot of volatile compounds that differed significantly or showed a tendency for difference (*p* < 0.1) among the seabream fillets, with consumer perceived freshness attributes presented as supplementary data.

**Figure 3 foods-12-00505-f003:**
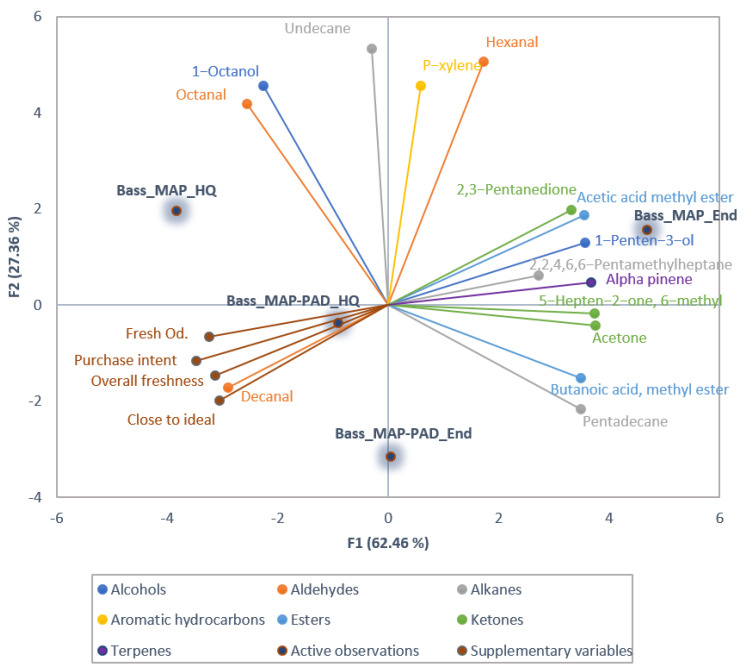
Principal component analysis (PCA) biplot of volatile compounds that differed significantly or showed a tendency for difference (*p* < 0.1) among the seabass fillets, with consumer perceived freshness attributes presented as supplementary data.

**Figure 4 foods-12-00505-f004:**
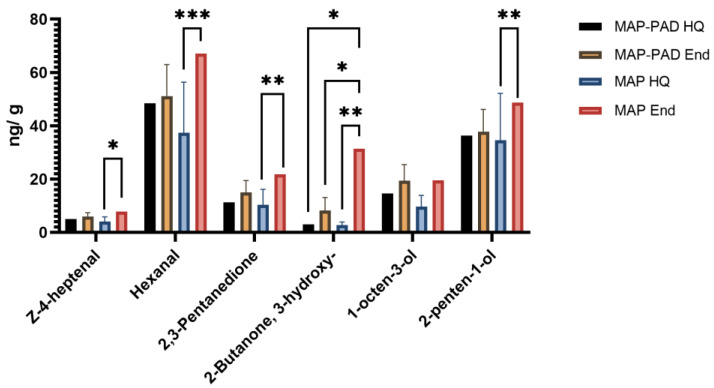
Concentration (ng/g) of target compounds identified via HS-SPME-GC-MS analysis in seabream fillets, packed in active packaging with (MAP-PAD) or without (MAP) a pad, at the high quality (HQ) time point or the end of high-quality (End) shelf-life. (*) stands for *p* < 0.05; (**) stands for *p* < 0.01; (***) stands for *p* < 0.001.

**Figure 5 foods-12-00505-f005:**
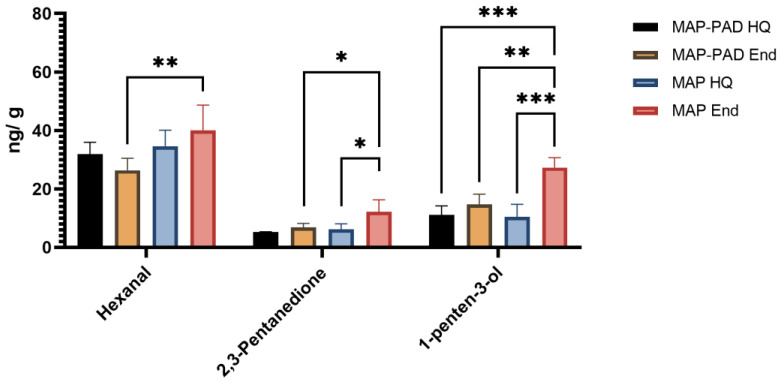
Concentration (ng/g) of target compounds identified via HS-SPME-GC-MS analysis in seabass fillets, packed in active packaging with (MAP-PAD) or without (MAP) a pad, at the high quality (HQ) time point or the end of high-quality (End) shelf-life. (*) stands for *p* < 0.05; (**) stands for *p* < 0.01; (***) stands for *p* < 0.001.

**Table 1 foods-12-00505-t001:** Questions asked in the consumer study for the whole fish and the fish fillets. Anchors and type of scale are also indicated.

Category	Question	Questionnaire Anchors for Whole Fish	Questionnaire Anchors for Fish Fillets	Scale
Sensory perception of raw samples	Odor	Very fresh to unpleasant	Very fresh to unpleasant	5-point
	Skin	Very bright, bright, dull	Very bright, bright, dull	3-point
	Eyes	Convex and shiny, less convex to flat and slightly dull, less convex to flat and dull	N/A *	3-point
	Liquid in the packaging	Presence or absence	Presence or absence	Binary
	Mucus	Thin and transparent or no mucus/dry	N/A	Binary
	Flesh	Very firm, elastic, fingerprint with pressure	Bright white and firm, white to dull and less firm, yellowish and soft	3-point
Overall perception of raw samples	Overall freshness	Very fresh to not fresh at all	Very fresh to not fresh at all	5-point
	Purchase intention	Very likely to not likely at all	Very likely to not likely at all	5-point
	Proximity to the ideal	Very close to not close at all	Very close to not close at all	5-point
Preference for raw samples	Which of the two samples do you prefer?			
Sensory perception of cooked samples	Odor	N/A	Very fresh to unpleasant	5-point
	Flavor	N/A	Pleasant/fresh, relatively fresh, neutral	3-point
	Juiciness	N/A	Juicy, slightly dry, dry	3-point
	Texture	N/A	Elastic/firm, soft	Binary
Overall perception of cooked samples	Liking	N/A	“I like it very much” to “I don’t like it at all”	7-point
	Overall freshness	N/A	Very fresh to not fresh at all	5-point

* N/A stands for not applicable.

**Table 2 foods-12-00505-t002:** Consumption frequency and attitudes towards fish and fish packaging. Results are presented as frequencies (*n*) and proportion (%) out of 274 consumers in total. *P*-values were calculated by applying the chi-square test at a = 0.05.

Questions	*n*	Proportion	*p*-Value
**Consumption frequency**			<0.001
Never	3	0.5%	
Rarely	2	0.5%	
Once to 3 times a month	92	34%	
Once or twice a week	177	65%	
**I usually buy: ***			<0.001
Fresh fish	261	95%	
Frozen fish	88	32%	
Canned fish	64	23%	
Other	7	3%	
**When I buy fresh fish, I usually buy it packaged.**	33	12%	<0.001
**I believe that packaged fish is: ***			<0.001
Less safe	36	13%	
Safer	104	38%	
Less convenient	23	8%	
More convenient	161	59%	
Less nutritious	34	12%	
More nutritious	8	3%	
Less fresh	137	50%	
Fresher	18	7%	
**Would you prefer a packaging with a pad?**			<0.001
Indifferent	84	31%	
No	39	14%	
Yes	151	55%	

* Participants were free to choose more than one answer for the respective questions.

**Table 3 foods-12-00505-t003:** Consumer preference results presented as frequencies (*n*). *p*-values were calculated by applying a binomial test at a = 0.05.

Fish Processing	Species	Shelf-Life Time Point	MAP-PAD	MAP	*p*-Value
Whole fish	Seabream	HQ	42	28	0.120
		End	54	12	<0.001
	Seabass	HQ	41	30	0.235
		End	40	27	0.142
Fish fillets	Seabream	HQ	38	24	0.098
		End	44	20	0.004
	Seabass	HQ	40	23	0.043
		End	48	14	<0.001

## Data Availability

The data are available from the corresponding author.

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
