# Peer review of "Consumer Perception of Freshness and Volatile Composition of Fresh Gilthead Seabream and Seabass in Active Packaging with and without CO2-Emitting Pads"

_foods, 2023, doi:10.3390/foods12030505_

Round 1

Reviewer 1 Report

The article entitled “consumer perception of freshness and volatile composition of fresh gilthead seabream and seabass in active packaging with and without CO2-emitting pads” was written well. 

Line 62, The “=“ seems incorrectly written

Line 75. HCMR was written. The abbreviation should be extended in the first time mentioned. 

In the sample preparation, the packaging concept was prepared for consumer test. Is the packaging system taken from other studies or already applied to one of industries in Greece? It is better to write the reference why the packaging system used for this study. 

It is suggested for author to mention the time for the end of shelf life of the packaged food. Each product with different packaging has different shelf life. So it is better clarify the shelf life as well

In Table 3. it is still confused how the data is presented. Are the numbers under MAP-PAD and MAP a frequency of consumer preference? if yes, why do they have different total for each analysis? For example seabream HQ was 42+28=70, but for the end the total was 54+12=66. Can author clarify this?

The supplementary of materials cannot be open. The link mentioned cannot be accessed. 

It is a raised question, how is the end of high quality determined? what is the indicator the products not high quality anymore? 

Author Response

First of all, we would like to thank Reviewer 1 for the thorough and constructive feedback. We have carefully revised the paper according to the feedback we received and now consider the manuscript to be improved according to the lines of the reviewer’s comments.

Please find in the attached document our responses to the comments of Reviewer 1. Responses are written in black, while reviewer’s comments are copied and pasted as such in blue color. Line numbers of the reviewer’s comments refer to the previously submitted version. Line numbers in the responses refer to the revised manuscript. We have used the “Track changes” function of MS Word to indicate the changes in the revised text. Other corrections and additions in the manuscript concern the comments of the rest of the reviewers.

Reviewer 2 Report

Manuscript MPB-D-22-01194R1: Consumer perception of freshness and volatile composition of 2 fresh gilthead seabream and seabass in active packaging with and without CO2-emitting pads

 2.1. Sample preparation and storage

Line 65: Fillets were skin-off or skin-on? From data on Table 1 they were skin-on. This information could be clearly indicated in this section

In the text it is not indicated how much time elapsed between slaughtering and packaging of the whole gutted fish and fillets. It is only in the Figure 1?

Additionally, information about the weight of fish/fillets should be indicated

Line 76: The definition of HCMR has to be indicated. The reviewer supposes that is Hellenic Centre for Marine Research.

Samples were kept at 2.5 °C. Why this temperature was selected since it is not the usual commercial temperature?

Line 90: It is not mentioned that the liquid means the presence of liquid in the packaging. Please rephrase.

Line 90: It is important that the authors clarify what they mean by coherent!

Line 109: The end of high-quality has been defined based on previous experience of MAP storage of the studied species.

The assumptions/reasons should be indicated in this article, making reference to the previous work, so as not to oblige the reader to consult that article.

Line 224: Finally, the participants of our study were asked, if they would prefer to buy a packaging with a pad over one without a pad and…Please rephrase the sentence.

Line 264: The number of panellists preferring the products at the end of storage (Table 3) was higher than the initial number in MAP-PAD. Is this true?

Line 272: Chi-square test analysis showed that, regardless the freshness stage, MAP-PAD conditions exhibited less liquid (p<0.05) than the respective MAP samples (Supplementary Table S4). This was expected since the liquid released is absorbed by the PAD. Thus, this statistical treatment is not very relevant.

Line 447: In whole-gutted fish MAP packages CO2 concentration ranged from 19% to 22% at the end of high-quality…In the section material and methods, it is not mentioned how the percentage of CO2 is measured. Please include such information.

 Author Response

First of all, we would like to thank Reviewer 2 for the thorough and constructive feedback. We have carefully revised the paper according to the feedback we received and now consider the manuscript to be more focused and precise.

Please find in the following, the authors’ responses to the comments of Reviewer 2. Responses are written in black, while reviewer’s comments are copied and pasted as such in blue color. Line numbers of the reviewer’s comments refer to the previously submitted version. Line numbers in the responses refer to the revised manuscript. We have used the “Track changes” function of MS Word to indicate the changes in the revised text. Other corrections and additions in the manuscript concern the comments of the rest of the reviewers.

Round 2

Reviewer 1 Report

The comments are well-addressed.